# Knowledge, Attitudes, and Practices of Iraqi Parents Regarding Antibiotic Use in Children and the Implications

**DOI:** 10.3390/antibiotics14040376

**Published:** 2025-04-03

**Authors:** Omeed Darweesh, Amanj Kurdi, Marwan Merkhan, Hemn Ahmed, Salih Ibrahem, Radhwan N. Al-Zidan, Johanna C. Meyer, Brian Godman

**Affiliations:** 1College of Pharmacy, Al-Kitab University, Kirkuk 36015, Iraq; amanj.baker@strath.ac.uk; 2College of Pharmacy, Hawler Medical University, Erbil 44001, Kurdistan Region, Iraq; 3Strathclyde Institute of Pharmacy and Biomedical Sciences, Strathclyde University, Glasgow G4 0RE, UK; 4Department of Public Health Pharmacy and Management, School of Pharmacy, Sefako Makgatho Health Sciences University, Pretoria 0208, South Africa; hannelie.meyer@smu.ac.za; 5College of Pharmacy, University of Mosul, Mosul 41002, Iraq; marwanmerkhan@uomosul.edu.iq (M.M.); radhwan.alzidan@uomosul.edu.iq (R.N.A.-Z.); 6Department of Pharmacy, Al-Qalam University College, Kirkuk 36001, Iraq; hemn.omer@alqalam.edu.iq; 7College of Dentistry, Kirkuk University, Kirkuk 36001, Iraq; salih.ibrahem@uokirkuk.edu.iq; 8South African Vaccination and Immunisation Centre, Sefako Makgatho Health Sciences University, Ga-Rankuwa 0208, South Africa; 9Antibiotic Policy Group, Institute for Infection and Immunity, City St George’s, University of London, London SW17 0RE, UK

**Keywords:** antibiotic misuse, antimicrobial resistance, KAPs, self-medication, community pharmacists, Iraq

## Abstract

**Background:** Antibiotic misuse is a major global health issue, particularly in low- and middle-income countries (LMICs), where inappropriate use contributes to antimicrobial resistance (AMR). Inappropriate antibiotic use is exacerbated in LMICs by pressure from parents on physicians and pharmacists to prescribe and dispense antibiotics for their children, often for viral infections. There is currently limited knowledge of key issues in Iraq to improve future antibiotic use. Consequently, we explored knowledge, attitudes, and practices (KAPs) of Iraqi parents regarding antibiotic use in their children to provide future direction. **Methods:** A cross-sectional survey was conducted between November 2023 and September 2024 among 1878 parents in four districts of Iraq. A structured questionnaire assessed parents’ KAPs towards antibiotic-related uses and practices. Data were analyzed using descriptive statistics and Chi-square tests to examine the associations between demographic variables and KAP indicators. **Results:** Among the participants, 83% were aware that antibiotics are ineffective for viral infections, and 75% recognized potential side effects from antibiotics. Despite this knowledge, 63% of parents admitted to administering antibiotics to their children without a prescription, with 42% discontinuing treatment once symptoms improved. Factors including cost, dosage, and taste influenced antibiotic selection. Parents with higher education and income levels were more likely to engage in self-medication. Pharmacists were the primary source of antibiotic information for 52% of respondents. **Conclusions:** Despite adequate knowledge, inappropriate practices such as self-medication and discontinuing treatment early were prevalent. Targeted educational campaigns, particularly among parents with higher education levels and incomes, are necessary to mitigate AMR in Iraq.

## 1. Introduction

Antibiotic misuse is a critical global health concern leading to antimicrobial resistance (AMR), with significant implications for public health, including among children [1,2]. According to the World Health Organization (WHO), AMR is now a significant public health concern, with appreciable morbidity, mortality and costs [2,3,4]. This is particularly the case among low- and middle-income countries (LMICs) [5,6]. The overuse and inappropriate use of antibiotics contributes to the development of antibiotic-resistant microbes, particularly those from the WHO Watch group [7,8,9]. Antibiotics are frequently misused in treating childhood illnesses, including for self-limiting conditions such as upper respiratory tract infections (URTIs), exacerbated by a lack of knowledge and inappropriate practices among parents [10,11,12]. This includes parents putting pressure on healthcare professionals (HCPs) to prescribe or dispense antibiotics for a range of conditions, which includes acute respiratory infections and acute diarrhea [12,13,14,15]. Alongside this, failure to complete the prescribed course of antibiotics due to a number of issues, which includes affordability and limited knowledge, especially among patients and parents in LMICs [12,16,17,18,19], has been observed.

The WHO and others have now instigated a range of activities and measures to address the rising rates of AMR [20,21,22]. This incorporates the WHO Global Action Plan, translating in National Action Plans [20,23,24,25]. Iraq is no exception, with ongoing monitoring of AMR and associated activities as part of its NAP [26,27]. Alongside this, the WHO has been involved in the development of the AWaRe (Access, Watch and Reserve) classification as part of updating the Essential Medicines List alongside the accompanying WHO AWaRE book, which gives treatment guidance for 35 infections to reduce the use of ‘Watch’ and ‘Reserve’ antibiotics, which display greater resistance potential [28,29,30,31]. This is critical in LMICs as the use of Watch and Reserve antibiotics is growing as a percentage of total antibiotic use in these countries, increasing AMR rates, and they currently have the greatest burden of AMR [32,33]. More recently, the United Nations General Assembly (UN GA) for AMR set ambitious targets for reducing AMR in view of its appreciable impact, which included enhancing the use of Access antibiotics, with AMR described by some as the next pandemic unless urgent actions are taken [34,35,36]. The WHO emphasizes the importance of appropriate use of antibiotics among the general population to control antibiotic misuse [37].

There are concerns with rising rates of AMR in Iraq, exacerbated by a series of conflicts in recent years and the corresponding impact on the healthcare system in the country [38]. In 2019, there were 3400 deaths attributable to AMR and 12,400 deaths associated with AMR in the country, with AMR being one of the leading causes of death in the country [39]. This is not helped by the fact that in Iraq, as in many other developing countries, there is considerable purchasing of antibiotics without a prescription [40,41,42].

The misuse of antibiotics is a complex issue influenced by a variety of factors, incorporating cultural and behavioral characteristics, including self-medication for mild childhood infections, socioeconomic factors, and educational aspects [41,42,43]. As a result of the socio-economic and cultural factors among LMICs, children are subjected to the misuse and overuse of antibiotics with oral antibiotics, enhanced by these being easily accessible without a prescription, including those from the Watch and Reserve list [44,45,46,47,48,49,50]. Parents’ misuse of antibiotics not only contributes to AMR but also impacts the broader community. This is a concern as the reduced effectiveness of antibiotics can lead to prolonged illness, increased healthcare costs, and a higher risk of complications [51,52].

However, we are aware there is currently an appreciable evidence gap regarding the use of antibiotics among children in Iraq and the possible key drivers for any misuse potentially driving up AMR in the country. Consequently, as a starting point, we aim to identify the gaps in knowledge, attitudes, and practices (KAPs) of antibiotic misuse among parents with children, especially as antibiotics are one of the most used class of medicines in children [53,54]. This builds on the publications of Al-Yasseri et al. (2019), Al-Taie et al. (2021) and Alsayed et al. (2022) [19,55,56]. Alongside this, we wanted to evaluate parents’ understanding of antibiotic indications, side effects, and AMR. The knowledge gained from this study, combined with pertinent findings from other countries, should help key stakeholder groups across Iraq develop targeted interventions, including targeted educational campaigns, to reduce antibiotic misuse in this vulnerable population in the country. This is important to achieve the NAP and UN GA goals in Iraq, especially as we are unaware of published studies assessing the KAPs of parents in Iraq since the publication of the NAP.

## 2. Results

### 2.1. Parent Demographics

A total of 1878 parents participated in this study. The majority of the participants were male (54%), and most were middle-aged, with 33% aged between 30 and 39 years, and 24% aged 40–49 years. The participants were distributed between Kirkuk, Erbil, Tikrit, and Mosul, with each city representing 25% of the total sample. Regarding education, 62% of the parents were university graduates, and 24% had a medical background. Most families reported a moderate income, with 46% earning between IQD 500,000 and 1,000,000 monthly (equivalent to ~USD 380–764), and 25.5% earning IQD > 1,000,000–1,500,000 (equivalent to ~>USD 764–1146). Additionally, 75% of parents had more than one child (Table 1).

### 2.2. Knowledge of Antibiotics

The knowledge section revealed that most parents (83%) understood that viral infections, such as the common cold, do not require antibiotics. Similarly, 83% knew that antibiotics are used to treat bacterial infections, and 75% were aware that antibiotic misuse can cause side effects, including diarrhea or allergic reactions. Awareness was notably higher among university graduates, males, and parents with medical backgrounds. Conversely, parents with lower educational levels and females were less knowledgeable about antibiotic risks (Table 2).

### 2.3. Attitudes Toward Antibiotic Use

The attitude indicators showed mixed results. While 67% of parents believed that only physicians should prescribe antibiotics, a significant portion (59%) expressed a preference for administering antibiotics to their children rather than waiting for symptoms to resolve naturally. Furthermore, 46% of parents believed that previously prescribed antibiotics could be reused for future illnesses with similar symptoms, and 64% admitted to storing leftover antibiotics at home for emergency use.

However, 55% of parents reported feeling satisfied when a physician opted not to prescribe antibiotics for a common cold or influenza symptoms. These attitudes varied significantly based on education and income, with parents from higher educational and income backgrounds showing greater restraint in using antibiotics unnecessarily (Table 3).

The “Agree” responses for “The physician is the only healthcare provider who should prescribe antibiotics” were significantly (*p* < 0.05) higher in university graduates’ high-income families compared to school graduates and low-income families, particularly when the graduates were from medical backgrounds. The “disagree” response of antibiotic re-use for similar symptoms was significantly (*p* < 0.05) higher in university graduates’ high-income families compared to school graduates and low-income families, particularly when the graduates were from medical fields.

The “agree” response for commencing antibiotic use rather than waiting until the symptoms spontaneously resolved was significantly (*p* < 0.05) higher in school graduates and low-income families compared to university graduates and high-income families, particularly when the graduates were from non-medical fields. The “agree” response for commencing antibiotic use only until the symptoms disappear was significantly (*p* < 0.05) higher in school graduates from low-income families compared to university graduates and high-income families, particularly when the graduates were from non-medical fields. The “agree” response for antibiotic-leftover use for emergencies was significantly (*p* < 0.05) higher in school graduates’ income families compared to university graduates and high-income families, particularly when the graduates were from non-medical fields. The “agree” response for parents being pleased with non-antibiotic prescriptions was significantly (*p* < 0.05) higher in low-income families compared to high-income families regardless of educational levels (Table 3).

### 2.4. Practices Surrounding Antibiotic Use

When examining actual practices, 63% of parents reported administering antibiotics to their children without a prescription. This behavior was more common among university graduates and parents from higher-income families. Of those who gave antibiotics, 51% did so to treat illnesses, while 19% used antibiotics as a preventive measure. The remaining 30% believed antibiotics could serve both therapeutic and prophylactic purposes. Regarding factors influencing antibiotic selection, 43% of parents prioritized cost, while 41% considered dosage frequency and duration of treatment. Notably, 17% based their choice on the antibiotic’s taste (Table 4). In terms of adherence, 42% of parents stopped the antibiotic treatment once their child’s symptoms improved, whereas 58% continued the full course as prescribed. The decision to continue the course was more common among university graduates and parents from higher-income households. Only 17% of parents reported giving their children higher-than-prescribed doses, and 82% refrained from using more than one antibiotic simultaneously without medical advice (Table 4).

The primary factor influencing parents’ selection of a specific antibiotic was the cost of the medication. However, 41% of parents also considered the number of daily doses and the duration of the treatment course when making their choice. Additionally, 17% of parents based their antibiotic selection on the taste of the medication. Once their children showed signs of recovery, 42% of parents reported discontinuing antibiotic use, while 58% indicated that they continued the full course of therapy. Notably, the majority of parents (83%) adhered to the prescribed doses and refrained from administering higher doses. Similarly, 82% of parents avoided using combined antibiotic therapies.

Stratification of the results based on demographic factors revealed significant associations with education level and family income. Parents who administered antibiotics to their children without a prescription were predominantly university graduates, medical professionals, and from higher-income families. Those who continued antibiotic therapy after symptom resolution were also more likely to be university graduates and medical professionals. While cost was a major determinant of antibiotic selection among lower-income families, higher-income families and medical professionals were more likely to base their choice on dosing schedules and treatment duration. Furthermore, parents who adhered to the full course of antibiotic treatment and avoided administering extra doses were primarily university graduates and medical professionals (Table 4).

### 2.5. Sources of Antibiotic Information

Pharmacists were the primary source of information for parents regarding antibiotic use, cited by 52% of participants. Other notable sources of information are included in Table 5. The general patterns of antibiotic misuse among parents revealed that one-quarter of parents obtained antibiotics without a prescription. The reasons for not consulting healthcare providers were varied, with approximately half of the parents believing that their child’s condition was not severe enough to warrant medical attention. A quarter of the parents felt confident in their ability to select the appropriate medication based on prior experience. In contrast, time constraints and financial limitations were less frequently cited as reasons for not seeking clinical consultation. The primary conditions for which parents initiated antibiotic use included fever, diarrhea, cough, and the common cold. Pharmacists were the most commonly relied-upon source of information to treat their children, consulted by half of the parents (Table 5).

## 3. Discussion and the Implications

We believe this is the first large and multicity study conducted in Iraq to assess parents’ KAPs relating to antibiotic use for their children since the publication of the Iraq NAP. This is important in this vulnerable population, especially given previous concerns regarding knowledge and use of antibiotics in ambulatory care in Iraq [19,55,56].

This study revealed that 83% of parents were aware that antibiotics are ineffective for treating viral infections such as the common cold. This is an encouraging and mirrors the findings in other LMICs [12,57]; however, different from other LMICs was that there were low levels of knowledge of antibiotics and AMR among parents [44,58,59]. Alongside this, we have also seen concerns where a good knowledge of antibiotics and AMR among parents has not translated into practice when parents treat their children with self-limiting conditions such as URTIs or diarrhea with antibiotics [12,60,61,62]. Encouragingly, this did not appear to be the situation in our study. However, discrepancies in knowledge based on gender and education levels were significant. Parents in our study who were university graduates and males, particularly those with medical backgrounds, demonstrated superior knowledge compared to females and those with lower educational levels. These findings are consistent with previous research among LMICs that has shown education plays a critical role in parents’ understanding of antibiotics and AMR [12,44,63,64,65]. This, though, is not always the case [66]. The association between educational levels and knowledge is noteworthy going forward when health authorities and others are preparing targeted campaigns to address misinformation and knowledge gaps among parents and the public to improve future antibiotic use. Encouragingly, parents with higher education levels demonstrated better awareness of appropriate antibiotic use, a trend also seen among a number of African and Asian countries [12,59,65,67,68,69,70]. This suggests that educational interventions and campaigns targeted especially at parents with lower levels of formal education may be particularly effective in improving antibiotic-related knowledge and practices. This reflects successful educational campaigns among the public across LMICs, including among pediatric caregivers [71,72,73,74,75]. However, care is needed regarding language with patients to avoid any misunderstanding of terminology surrounding antibiotics and AMR [76,77,78,79].

Nonetheless, significant gaps remain, particularly in recognizing the adverse effects of antibiotics. In our study, 25% of parents were unaware of the risks, including side effects such as allergic reactions and diarrhea, evidenced by 9% disagreeing that antibiotics can be associated with side effects and 16% being unsure. Similar knowledge gaps regarding antibiotic risks have also been documented in other LMICs, indicating that while basic knowledge exists, understanding of the consequences of misuse is often suboptimal in these countries [12,80,81].

Another identified key concern was that despite relatively high levels of knowledge about antibiotics and AMR, with 67% of surveyed parents believing that antibiotics should only be prescribed by a physician, 59% of participating parents expressed a preference for administering antibiotics to their children rather than waiting for symptoms to resolve naturally when they are self-limiting. This behavior is concerning as it perpetuates antibiotic misuse and associated AMR. However, this reflects similar attitudes in other LMICs, where antibiotics are often perceived as a quick solution for treating infectious diseases regardless of their appropriateness [12,51,82,83,84]. The belief that antibiotics are necessary for rapid recovery, even for self-limiting viral infections, remains a significant barrier to appropriate antibiotic use across LMICs, which needs to be addressed going forward [12,83,84,85]. The findings that parents from lower-income families, and those with less education, were more likely to favor immediate antibiotic use again underscores the need for public health messaging initially among this parent group in Iraq to help reduce unnecessary antibiotic use and AMR. Interestingly, our findings revealed that university-educated and higher-income parents were more likely to engage in self-medication with antibiotics for their children, a pattern also observed in other LMICs [12,42]. While higher education is often associated with greater health literacy, research indicates that knowledge alone does not consistently translate into appropriate health behaviors. Behavioral choices, particularly around medicine use, are influenced by a complex interplay of cognitive, social, and contextual factors, including attitudes, norms, convenience, and risk perception [86]. One explanation may be that better-educated individuals feel more confident in their ability to self-diagnose and treat perceived minor infections without consulting an HCP. This sense of autonomy and overconfidence has been previously reported, including in a cross-sectional study in China where higher parental education was associated with increased antibiotic self-medication in children [12,66,87,88]. Other studies have also shown that increased education sometimes correlates with a higher propensity for self-medication [66,89,90]. Alongside this, higher-income families may have greater access to antibiotics. This may be through greater purchasing power, social networks, or using leftover medicines from previous prescriptions, thereby enhancing their non-prescribed use. These findings highlight the importance of designing behaviorally informed interventions that go beyond knowledge dissemination and explicitly address attitudes, perceived capabilities, and access-related enablers of inappropriate antibiotic use [12,91].

Moreover, nearly half of the surveyed parents believed that previously prescribed antibiotics would be effective and could be reused for future illnesses with similar symptoms. This practice, driven by both economic factors and misconceptions about antibiotics and their effectiveness including for viral infections, has been well documented across LMICs and again contributes to AMR [12,56,92,93,94,95]. The retention, sharing and future use of leftover antibiotics (reported by 64% of parents) further complicate efforts to curb inappropriate antibiotic use in ambulatory care, increasing AMR, with similar findings across LMICs [95,96,97,98].

In terms of practices, this study found that 63% of parents admitted to giving antibiotics to their children without a prescription, highlighting the persistence of self-medication despite widespread knowledge about the dangers of antibiotic misuse. This high rate of self-medication mirrors findings from studies in other LMICs, including among African and Asian countries, where weak regulatory enforcement and cultural norms contribute to the widespread availability and use of antibiotics without professional oversight [41,42,46,47,81,90,99,100,101]. Interestingly, as mentioned, parents with higher educational levels and incomes were more likely to administer antibiotics without a prescription, perhaps reflecting overconfidence in their medical knowledge [88,89,90].

Stopping antibiotic treatment prematurely, reported by 42% of parents in this study, and selecting antibiotics based on cost, as noted by 43% of parents, are practices that significantly contribute to the development of antimicrobial resistance (AMR). Incomplete antibiotic courses allow bacteria to survive and mutate, leading to resistant strains, a trend observed not only in Iraq but also across LMICs [102,103,104,105,106]. Similarly, the choice of antibiotics based on cost reflects the economic realities faced by families across LMICs, where financial constraints influence healthcare decisions [12,81,99,100].

### Implications for Clinical Practice, Policy-Makers, and Public Health in Iraq

The findings from this study have significant implications for clinical practice, policy-making, and patients in Iraq, and provide insight related to achieving reductions in AMR in line with the goals of the NAP. Alongside this, appropriate prescribing and dispensing of antibiotics should be enhanced in order to achieve the UN GA goals set out in September 2024, which includes greater use of Access antibiotics where antibiotics are indicated [36]. Suggested activities among key stakeholder groups are contained in Table 6. Furthermore, interprofessional education (IPE) and enhanced communication among HCPs are pivotal in promoting appropriate antibiotic use and combating AMR. Collaborative practice, particularly between physicians and pharmacists, ensures the consistent delivery of evidence-based information to parents regarding the risks associated with antibiotic misuse and the importance of adherence to prescribed regimens. The WHO underscores that effective IPE fosters respect among health professions, enhances communication skills, and promotes collaborative practice, all of which are essential in addressing complex health challenges, including rising AMR [107]. In the context of Iraq, recent studies indicate a suboptimal level of interprofessional collaboration among healthcare team members. To illustrate this, a study conducted in critical care units in Najaf Governorate revealed poor overall interprofessional collaboration, with weak scores in partnership (mean score: 2.31) and coordination (mean score: 2.17), and only a moderate score in cooperation (mean score: 2.51) [108]. The study concluded that the overall interprofessional collaboration among healthcare team members was substandard, highlighting the need for enhanced interprofessional education and communication strategies within the Iraqi healthcare system [108]. Given that pharmacists were identified as the primary source of information for over half of the parents in this study, implementing shared learning initiatives and coordinated antimicrobial stewardship (AMS) interventions, particularly at the primary care and community pharmacy levels, could significantly enhance public understanding and adherence to antibiotic guidelines. Community pharmacists played a key role in the COVD-19 pandemic providing information to help prevent the spread, as well as treat patients, and this trend should continue [109,110,111]. Integrating IPE principles into both undergraduate training and continuing professional development programs is essential to foster future collaboration and develop a unified response to combat AMR in Iraq and other LMICs.

We are aware of a number of limitations with this study. As with all self-reported survey studies, this research is subject to potential information bias, particularly self-reporting and social desirability bias, which may have led some participants to under-report inappropriate behaviors, such as self-medicating or prematurely stopping antibiotics, or to overstate practices perceived as socially acceptable. These biases could affect the accuracy of the reported prevalence of antibiotic misuse. To mitigate these risks, several strategies were implemented. Firstly, participants were assured of the anonymity and confidentiality of their responses. Secondly, the questionnaire was developed based on previously validated instruments, and, lastly, interviews were conducted face-to-face to allow for the clarification of questions while minimizing interviewer influence. Despite these efforts, we acknowledge that some degree of bias may persist. This is a recognized limitation of KAP studies relying on self-report and should be considered when interpreting the findings [133]. Alongside this, the survey underwent piloting and refinement, allowing for the clarification of ambiguous items and the use of neutral phrasing, which may have helped minimize biased reporting. Nonetheless, the possibility of residual bias cannot be entirely excluded.

Overall, we believe that this study possesses several key strengths that enhance its contribution to the existing literature to help improve antibiotic use in children in Iraq and across LMICs. Firstly, it represents one of the largest and most comprehensive investigations regarding the KAPs of Iraqi parents concerning antibiotic use in their children. By surveying 1878 participants across four geographically and demographically diverse districts (Kirkuk, Erbil, Tikrit, and Mosul), this study ensures wide generalizability of its findings across different regions of Iraq. The inclusion of multiple districts also allows for a more representative sample, reflecting the variability in cultural, socioeconomic, and healthcare access factors influencing antibiotic use.

Secondly, we believe this is the first study undertaken in Iraq to assess KAPs on antibiotic use among Iraqi parents since the implementation of Iraq’s NAP on AMR. By identifying gaps in parental knowledge and inappropriate practices, this study provides critical baseline data for evaluating the impact of current and future public health interventions relating to antibiotic stewardship in the country.

Thirdly, this study employed a structured and validated survey instrument, designed based on the existing literature and refined through expert consultation. The use of face-to-face data collection ensured high response rates and data completeness, reducing potential biases associated with self-administered surveys. Moreover, statistical rigor was applied through stratified analyses, allowing for a deeper understanding of the role of demographic factors, e.g., education level, income, and medical background, in influencing parental behaviors towards antibiotic use. Finally, this study contributes to the global conversation on antibiotic misuse in LMICs by drawing parallels with findings from other countries and regions. The insights derived have implications beyond Iraq, serving as a valuable reference for other LMICs facing similar challenges in curbing inappropriate antibiotic use and tackling AMR.

While this study provides important insights into the associations between parental characteristics and patterns of antibiotic use, we acknowledge that future research could benefit from the application of multivariable modeling techniques, including logistic regression, to adjust for potential confounding variables and explore independent predictors of key behaviors. Such approaches would be especially valuable in studies aiming to establish causal inferences or develop targeted interventions for high-risk groups.

Overall, we believe the findings of our study offer a foundational understanding of antibiotic-related knowledge, attitudes, and practices among parents in Iraq and can serve as a basis for designing more complex analytical models in future investigations. The findings also support the tailoring of AMS interventions and public health messaging strategies to the specific behavioral drivers identified across different socioeconomic and educational strata. Furthermore, data collection was carried out as scheduled between November 2023 and September 2024 across all four study locations. There were no deviations from the planned timeline. Recruitment, data collection, and entry were completed within this period, ensuring consistency in the temporal context of responses and enhancing the internal validity of the findings.

## 4. Materials and Methods

### 4.1. Study Design and Population

A survey-based cross-sectional study was conducted in four Iraqi districts (covering around 9.4% of the total Iraqi population of ~46 million) between November 2023 and September 2024. Parents with at least one child were enrolled in this study and were provided with a printed copy of the questionnaire. These cities were purposively selected to ensure geographical and demographic diversity across northern and central Iraq. A stratified convenience sampling strategy was employed. Within each city, the participants were recruited from various high-footfall public locations such as shopping centers, parks, outpatient clinics, and educational institutions in order to capture a diverse cross-section of the population. Stratification was based on city of residence to ensure equal representation from each site (25% per city), while efforts were made during recruitment to balance enrolment across key sociodemographic strata, including education levels (high school or lower vs. university graduate or higher) and different income categories (IQD ≤ 500,000; IQD 500,000–1,000,000; IQD > 1,000,000).

The parents were informed about the aim of this study, and it was requested that only one parent from a family should answer the questionnaire (either the mother or the father). Participants with literacy difficulties were given the alternative of having the questions read to them and their answers were recorded by the principal author (OD). This was to minimize any discomfort they may have and increase their participation.

The questionnaire was not given to those who refused to participate. The sample size was calculated using the Raosoft program (http://www.raosoft.com/samplesize.html; accessed on 8 September 2024) taking into account the total population of these cities, which is 4,369,177 individuals. A minimum representative sample size of 664 was determined, based on a confidence interval of 99% and a margin of error of 5%. Nevertheless, the sample size was subsequently increased threefold to strengthen the statistical analysis. This expansion was implemented to enhance the statistical power in order to allow for more robust subgroup analyses across education, income, and professional background strata, as well as to improve the generalizability of the findings. We extended data collection beyond the minimum target. Recruitment continued until an approximately threefold increase was reached (*n* = 1878), using the same proportional allocation strategy across cities.

### 4.2. Instruments and Data Collection

We designed a knowledge, attitude, and practices (KAPs) questionnaire by reviewing the literature of previous KAP-type studies regarding antibiotic misuse [97,134], as well as consulting several researchers from Al-Kitab University-College of Pharmacy with expertise in clinical pharmacy, bacterial resistance, and epidemiology (Appendix A).

The study questionnaire was subsequently piloted with ten parents to add robustness to the questionnaire that was used in the main study. No major changes were made to the questionnaire, with data from the pilot study subsequently excluded from the main study.

A total of four sections were included in the questionnaire, with each section including multiple questions. The first section included the participants’ socio-demographic characteristics, such as their age, gender, education level, income level and socioeconomic status. The second section included questions assessing the participants’ understanding of antibiotics, their uses, their role in treating bacterial infections, self-prescription, side effects and resistance to bacteria. The third section included statements exploring the participants’ attitudes towards antibiotics, e.g., their beliefs about the effectiveness of antibiotics for various illnesses. In addition, questions about attitudes towards self-medication and whether participants believed it is acceptable to use leftover antibiotics without consulting a healthcare professional were included, as well as questions about sources of information regarding antibiotic use. The fourth section included questions about the participants’ actual practices relating to their antibiotic use, including whether they have taken antibiotics without a prescription, as well as questions about how participants obtain their antibiotics, whether from healthcare professionals, over-the-counter sources, or other means.

The questionnaire was designed to collect categorical, item-level responses across key domains of knowledge, attitudes, and practices related to antibiotic use. Each item was developed to function independently and analyzed accordingly, without aggregation into composite scales, rather than developing a scale-based instrument aimed at quantifying unidimensional latent constructs. Consequently, psychometric techniques including internal consistency testing using Cronbach’s alpha or construct validity assessment through factor analysis were not applicable as these techniques are only appropriate when items are designed to measure the same latent construct. Nevertheless, to ensure content and face validity, the questionnaire was developed based on previously validated KAP studies and refined through consultation with subject matter experts in clinical pharmacy, epidemiology, and public health. Furthermore, as mentioned earlier, a pilot test was conducted with a small group of parents to ensure clarity and relevance of all items. The final version of the questionnaire is included as Appendix A.

Family income level was categorized into four categories based on the families’ average monthly income.

Future guidance to all key stakeholder groups was based on the considerable experiences of the co-authors working across LMICs and providing similar advice to other key groups [9,10,25,135].

A Microsoft Excel spreadsheet was used to enter data received from participants while ensuring their anonymity. Data were collected through face-to-face interviews with the study participants.

### 4.3. Statistical Analysis

Data from this study were analyzed using GraphPad Prism (version 10). Descriptive statistics were used to summarize the participants’ demographic characteristics, as well as their KAPs regarding antibiotic use. Categorical variables were expressed as frequencies and percentages. The association between demographic variables (such as age, gender, educational level, and income) and the participants’ KAPs were assessed using the Chi-square test. Statistical significance was determined at a *p*-value of less than 0.05.

For comparative analyses, the participants were stratified based on educational level, income, and medical background to assess the impact of these variables on the study outcomes. Chi-square tests were employed to examine significant differences between groups for specific KAP indicators. Where relevant, a post hoc analysis was conducted to identify specific group differences. All results are presented in the form of tables to illustrate key findings and highlight statistically significant relationships between the demographic factors and parental behaviors regarding antibiotic use in children.

Given the exploratory nature of this study, we limited the analysis to Chi-square tests to examine associations between parental demographic characteristics and KAP indicators. Multivariable regression techniques, such as logistic regression, were not employed, as this study was not intended to produce predictive models but rather to identify broad, stratified patterns to inform future research and public health interventions. This analytical approach is consistent with similar exploratory KAP studies conducted in LMICs and supports clearer communication of findings to a broader audience, including public health stakeholders and policy-makers, as part of interprofessional teamwork [136,137]. The use of such bivariate methods in early-phase, observational studies is well established in applied health research [138].

## 5. Conclusions

Overall, this study highlights important challenges in antibiotic use among Iraqi parents, which mirror broader issues faced across LMICs, highlighting the need for targeted educational and other interventions to address antibiotic misuse among parents and the general public in Iraq. While knowledge levels are relatively high, significant gaps in attitudes and practices remain, particularly among less-educated and lower-income families, contributing to the growing threat of AMR in Iraq.

Targeted educational programs that focus on correcting misconceptions, especially about the use of antibiotics for viral infections, are crucial going forward. Policy-makers must also strengthen the regulatory frameworks to limit inappropriate dispensing of antibiotics, while public health systems should focus on improving access to affordable healthcare. Future research should explore the effectiveness of specific interventions, including community-based antibiotic stewardship programs and targeted educational campaigns, in changing inappropriate antibiotic use behavior to meet NAP and UN GA goals for AMR.

## Figures and Tables

**Table 1 antibiotics-14-00376-t001:** Demographic parameters of the studied groups (*n* = 1878).

Demographic Parameters	Types	*n* (%)
Age range (Years)	19–29	548 (29%)
	30–39	614 (33%)
	40–49	448 (24%)
	≥50	268 (14%)
Sex	Male	1011 (54%)
	Female	867 (46%)
Residency	Kirkuk	470 (25%)
	Erbil	470 (25%)
	Tikrit	469 (25%)
	Mosul	469 (25%)
Parents educational levels	High school or lower	721 (38%)
	University graduate or higher	1157 (62%)
Family income status (USD 1 = IQD 1500)	IQD ≤ 500,000	421 (22.5%)
	IQD 500,000–1,000,000	866 (46%)
	IQD > 1,000,000–1,500,000	481 (25.5%)
	IQD > 1,500,000	110 (6%)
Are any of the parents from the medical staff?	Yes	443 (24%)
	No	1435 (76%)
No. of children	1	378 (20%)
	2	775 (41%)
	3	640 (34%)
	4	85 (5%)

NB: IQD = Iraqi Dinar; USD = United State Dollar.

**Table 2 antibiotics-14-00376-t002:** Parents’ knowledge towards their children’s intake of antibiotics. Data expressed as frequency.

Knowledge Indicators	Frequency of Response *n* (%)		
	Agree	Disagree	Do Not Know
Most symptoms of cough, influenza and cold that affect children result from viral infection.	1555 (83%) U > S * M > F * PM > PnM *	145 (7.7%) S > U * F = M PnM > PM *	178(9.3%) S = U F = M PnM > PM *
Antibiotics are used to treat bacterial infections.	1559 (83%) U > S * M > F * PM > PnM *	123 (7%) S > U * F = M PnM > PM *	196 (10%) S = U F = M PnM > PM *
Antibiotic use may cause side effects such as diarrhea or allergies.	1415 (75%) U > S * M > F * PM > PnM *	169 (9%) S > U * F = M PnM > PM *	294 (16%) S = U F = M PnM > PM *
Giving antibiotics to the child without a prescription may expose them to side effects.	1431 (76%) U > S * M > F * PM > PnM *	250(13%) S > U * F = M PnM > PM *	197(11%) S = U F = M PnM > PM *

NB: S = School graduate; U = University graduate; M = male; F = female; PM = Parent medical background; PnM = Parent non-medical background; * = indicates significant differences at *p*-value (<0.05) using Chi-square.

**Table 3 antibiotics-14-00376-t003:** Parents’ attitude towards their children’s intake of antibiotics.

Attitude Indicators	Frequency of Response *n* (%)		
	Agree	Disagree	Do not Know
The physician is the only healthcare provider who should prescribe antibiotics	1256 (67%) U > S * PM > PnM * HIF > LIF *	549 (29%) S > U * PnM > PM * LIF > HIF *	73 (4%) S > U * PnM > PM * LIF = HIF
The antibiotic which I used after the physician’s prescription will be always effective in treating similar symptoms	859 (46%) S > U * PnM > PM * LIF > HIF *	808 (43%) U > S * PM > PnM * HIF > LIF *	211 (11%) S > U * PnM > PM * LIF = HIF
I prefer to give an antibiotic to my children rather than wait until they become better without it	1110 (59%) S > U * PnM > PM * LIF > HIF *	653 (35%) U > S * PM > PnM * HIF > LIF *	115 (6%) S > U * PnM > PM * LIF = HIF
Antibiotics are used until symptoms resolve	1480 (79%) S > U * PnM > PM * LIF > HIF *	279 (15%) U > S * PM > PnM * HIF > LIF *	119 (6%) S > U * PnM > PM * LIF = HIF
I keep antibiotics or leftover antibiotics at home for emergency conditions.	1197 (64%) S > U * PnM > PM * LIF > HIF *	574 (30%) U > S * PM > PnM * HIF > LIF *	107 (6%) S > U * PnM > PM * LIF = HIF
When a physician does not prescribe antibiotics for symptoms of a common cold, nasal congestion or flu for the child, you are pleased with his prescription	1031 (55%) LIF > HIF *	627 (33%) LIF = HIF	220 (12%) LIF = HIF

NB: S = school graduate; U = university graduate; PM = Parent medical background; PnM = Parent non-medical background; LIF = low-income family; HIF = high income family. * indicates significant differences at *p*-value (<0.05) using Chi-square.

**Table 4 antibiotics-14-00376-t004:** Parents’ practice of their children’s intake of antibiotics. Data expressed as frequency.

Practice Indicators	Answer	*n* (%)	
Did you give antibiotics to your child without a prescription previously?	Yes	1190 (63%)	U > S *, PM > PnM *, HIF > LIF *
	No	688 (37%)	
Parents gave antibiotics to:	Treat the disease	955 (51%)	U > S *, PM > PnM *
	Prevent disease occurrence	360 (19%)	S > U *, PnM > PM *
	Both answers	563 (30%)	S > U *, PnM > PM *
The main factor in choosing a specific type of antibiotic was	Requiring lower number of times daily	443 (24%)	U > S *, PM > PnM *, HIF > LIF *
	Requiring lower number of days	317 (17%)	U > S *, PM > PnM *, HIF > LIF *
	Acceptable taste	295 (16%)	U > S *, PM > PnM *, HIF > LIF *
	Cost	823 (43%)	LIF > HIF *
If your child was feeling well, did you stop your child’s medication before the end of the antibiotic-prescribed course?	Yes	789 (42%)	S > U *, PnM > PM *
	No	1089 (58%)	U > S *, PM > PnM *
Did you give your child an extra dose (more than the prescribed doses by the physician)?	Yes	322 (17%)	S > U *, PnM > PM *
	No	1556 (83%)	U = S, PM = PnM, HIF = LIF
Did you give your child more than one antibiotic at one time without medical advice (injection plus suspension)?	Yes	329 (18%)	Reasons (*n*) Allergy (*n* = 18) The child did not respond well (*n* = 57) Overcome resistance (*n* = 49) Prescribed (25) Faster recovery (150) One antibiotic ineffective (*n* = 30)
	No	1549 (82%)	

* indicates significant differences at *p*-value (<0.05) using Chi-square.

**Table 5 antibiotics-14-00376-t005:** Parents’ practice of their children’s intake of antibiotics. Data expressed as frequency (*n* = 1878).

Practice Indicators	Sources of antibiotics used	Purchased from the pharmacy with a prescription		Purchased from the pharmacy without prescription		Purchased from other places		Total Number of Responses *
		1430 (59.9%)		871 (36.5%)		87 (3.6%)		2388
	Reasons for not consulting the physicians	Symptoms are not severe and do not require physician consultation		I have previous experience with drug efficacy	Lack of time	Lack of money	Others	Total Number of Responses *
		1200 (48.6%)		770 (31.2%)	238 (9.6%)	162 (6.6%)	97 (4%)	2467
	Medical conditions for which antibiotics were used	Fever	Diarrhea	Cough and the common cold		Ear pain	Other conditions	Total Number of Responses *
		1430 (32.6%)	1029 (23.5%)	1225 (28%)		555 (12.7%)	142 (3.2%)	4381
	Source of information for antibiotic use	Pharmacist	Nurse	Medication packages	Friends	Social media	Previous experience	Total Number of Responses *
		1716 (51.7%)	416 (12.6%)	338 (10.2%)	132 (4%)	121 (3.7%)	589 (17.8%)	3312

NB: * Multiple responses were analyzed and gathered for each to reach the total frequency for each response.

**Table 6 antibiotics-14-00376-t006:** Suggested future activities among all key stakeholder groups in Iraq to improve future use of antibiotics and decrease AMR.

Stakeholder Group	Suggested Goals and Activities
Policy-Makers/Health Authorities/Public Health	Work with universities/curriculum personnel to ensure every new healthcare professional leaving university is fully cognizant with all key aspects of antibiotics and AMR, including the WHO AWaRe classification and guidance book, as well as the goals of the Iraq NAP and its implications, with increasing use of the WHO AWaRe classification to assess current antibiotic utilization across sectors [26,28,31,112,113]. As a result, healthcare professionals should be able to discuss the appropriate use of antibiotics with patients post qualification, as well as the rationale. Work with physicians and their associations across sectors to implement possible ASPs including monitoring prescribing practices against agreed practices and sectors and including against agreed quality indicators [114,115,116,117]. Alongside this, key policy-makers, health authorities and public health personnel need to work with community pharmacists and their organizations/governing bodies to improve the appropriateness of any antibiotics dispensed in the community, building on their educational role during pandemics including COVID-19 [109,110,118], as there have been concerns with appropriate medicine use in Iraq including during the pandemic [119]. We have seen among African and Asian countries that the presence of well-trained pharmacists reduces unnecessary dispensing of antibiotics, including without a prescription, for essentially viral infections, including during the COVID-19 pandemic [99,111,120,121,122]. Potential activities among health authority/public health personnel include monitoring the dispensing of antibiotics among community pharmacists through mobile telephone technologies and other IT approaches against agreed-upon guidance or indicators [12,43,123]. Countries such as the UK allow community pharmacists to dispense antibiotics without a prescription for targeted infections when discussing these with patients [124], with the potential for such initiatives to be rolled out in LMICs [99]. However, it should be ensured that current Drug Laws are appropriate for the situation, otherwise there can be concerns [125]. Re-evaluate current regulatory frameworks and activities to reduce inappropriate sales of antibiotics without a prescription through community pharmacists and other outlets. However, it should be ensured that any sanctions, including any fines, are appropriate for the situation given concerns and experiences among LMICs [99,126]. The goal is to ensure good access to affordable healthcare and antibiotics, with any prescribing or dispensing of antibiotics in accordance with robust recommendations, as seen in the WHO AWaRe guidance book [31,88,112].
Physicians and Community Pharmacists	The findings highlight several critical implications for both clinicians and community pharmacists in their roles as regards improving future antibiotic use and combating AMR in Iraq. Clinicians, particularly pediatricians, play a pivotal role in reinforcing proper antibiotic use. They should ensure that parents fully understand the risks associated with antibiotic misuse, including the dangers of self-medication and premature discontinuation of treatment. Emphasizing the importance of completing prescribed courses of antibiotics, and addressing misconceptions around antibiotic use for essentially viral infections are critical to reduce inappropriate antibiotic prescribing exacerbated by requests from patients [127,128,129,130]. Community pharmacists, as primary sources of antibiotic information for parents, are uniquely positioned to guide and educate on the proper use of antibiotics. With 52% of parents in this study relying on pharmacists for advice, pharmacists must actively engage in antimicrobial stewardship activities, ensuring antibiotics are dispensed only when appropriate. Community pharmacists should also educate parents about the risks of obtaining antibiotics without a prescription, and the importance of adhering to treatment regimens acknowledging that they must make sure patients are conversant with terms such as antibiotics and AMR [76,79,131,132]. Strengthening the collaboration between clinicians and pharmacists through shared responsibility in antibiotic stewardship will become increasingly important across countries, including Iraq, to reducing AMR and improve public health outcomes. Implementing regular training programs and reinforcing adherence to national guidelines can further equip both groups to tackle the issue effectively, especially given that the WHO AWaRe guidance published covers 35 infectious diseases commonly seen across sectors.
Patients and the Public	The findings also underscore the crucial role of patients, particularly parents, in preventing AMR by increasing their responsible antibiotic use. Targeted educational campaigns, especially those aimed at patients with lower levels of education, must convey that fact that not all infections, such as those caused by viruses, e.g., common colds and influenza, require antibiotics. In fact, inappropriate treatment can be both more costly and potentially cause harm. Patient organizations, as well as university academics involved with researching the use of antibiotics among patients, can help in this regard especially given likely limited resources for comprehensive educational programs. Educating parents about the risks associated with inappropriate antibiotic use, including the development of resistant bacteria and potential side effects, is essential. Parents should be encouraged where possible to seek medical advice before using antibiotics and to avoid self-medicating or purchasing antibiotics without a prescription. There is also a need to educate patients regarding the importance of completing the full course of antibiotics as prescribed, even if symptoms improve, to prevent the survival and mutation of bacteria into resistant strains. Physicians and community pharmacists are critical to address current knowledge concerns. Public health campaigns and community education programs should target patients, focusing on increasing awareness of when antibiotics are necessary and the dangers of misuse. By equipping patients with this knowledge, healthcare systems can reduce the prevalence of AMR and promote better health outcomes at both the individual and community levels. For the broader public, understanding the impact of AMR on health systems and individual well-being is essential. Public health initiatives should aim to create a culture of responsible antibiotic use, encouraging individuals to consult healthcare professionals before using antibiotics and to adhere strictly to prescribed treatments. By raising awareness and fostering behavior change, these combined activities can reduce antibiotic misuse and help safeguard community health, ultimately improving public health outcomes and reducing healthcare costs.

## Data Availability

Additional data are available from the co-authors following reasonable requests.

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
