# Peer review of "Knowledge, Attitudes, and Practices of Iraqi Parents Regarding Antibiotic Use in Children and the Implications"

_antibiotics, 2025, doi:10.3390/antibiotics14040376_

Round 1
Reviewer 1 Report
Comments and Suggestions for Authors
The authors aimed to understand the relationships between the knowledge, attitudes, and practices (KAP) of Iraqi parents and the inappropriate use of antibiotics, in addition to parents' understanding of antibiotic indications, adverse effects, and antimicrobial resistance (AMR).
The introduction adequately outlines the problems and concerns regarding the misuse of antibiotics, and the goal of the study is clearly stated. The authors have provided sufficient background information.
The study design and data collection methods are well described. The questionnaire is detailed enough to allow for replication.
The authors present the findings logically and clearly. Tables and figures are used effectively to illustrate the key findings of the study.
The authors discuss the results in the context of current knowledge and existing literature. The authors also address the discrepancies between knowledge and practice and suggest practical solutions, such as targeted educational campaigns, to reduce antibiotic misuse in vulnerable populations in Iraq. The implications for future research and practice are also discussed.
Minor comments:
Would the authors be able to comment on interprofessional education and communication among health care professionals in the discussion?
Line 224: .... children since the publication of the Iraq NAP. Shoult "the" in front of Iraq be removed?
Author Response
Quality of English Language
( ) The English could be improved to more clearly express the research.
(x) The English is fine and does not require any improvement.
Author comments: Thank you – appreciated
A) Comments and Suggestions for Authors
a) The authors aimed to understand the relationships between the knowledge, attitudes, and practices (KAP) of Iraqi parents and the inappropriate use of antibiotics, in addition to parents' understanding of antibiotic indications, adverse effects, and antimicrobial resistance (AMR).
The introduction adequately outlines the problems and concerns regarding the misuse of antibiotics, and the goal of the study is clearly stated. The authors have provided sufficient background information.
Author comments: Thank you – appreciated
b) The study design and data collection methods are well described. The questionnaire is detailed enough to allow for replication.
Author comments: Thank you – appreciated
c) The authors present the findings logically and clearly. Tables and figures are used effectively to illustrate the key findings of the study. The authors discuss the results in the context of current knowledge and existing literature. The authors also address the discrepancies between knowledge and practice and suggest practical solutions, such as targeted educational campaigns, to reduce antibiotic misuse in vulnerable populations in Iraq. The implications for future research and practice are also discussed.
Author comments: Thank you – appreciated
B) Minor comments:
a) Would the authors be able to comment on interprofessional education and communication among health care professionals in the discussion?
Author comments: We thank the reviewer for this valuable suggestion. We have now added a paragraph to the discussion to address the role of interprofessional education and communication in supporting appropriate antibiotic use and mitigating AMR. The revised text emphasises how collaborative practice between physicians and pharmacists, supported by shared educational efforts, can help ensure consistent and evidence-based advice to parents, particularly in community settings to improve future antibiotic use. We trust this is now acceptable
b) Line 224: .... children since the publication of the Iraq NAP. Shoult "the" in front of Iraq be removed?
Author comments: Thank you – I believe it reads better to keep ‘the’ in front of Iraq. Hopefully, this is acceptable.

Reviewer 2 Report
Comments and Suggestions for Authors
Nowadays when antibiotic misuse is not only extremely high, but also has devastating effects by increasing antimicrobial resistance, such studies are helpful in creating global policies for the use of antibiotics.
The authors applied the questionnaire to a significant group of the population, so the results are relevant, although there are some limitations.
The references are numerous and current, including World Health Organization guides.
Author Response
Quality of English Language
( ) The English could be improved to more clearly express the research.
(x) The English is fine and does not require any improvement.
Author comments: Thank you – appreciated
a) Nowadays when antibiotic misuse is not only extremely high, but also has devastating effects by increasing antimicrobial resistance, such studies are helpful in creating global policies for the use of antibiotics.
Author comments: Thank you – appreciated
b)The authors applied the questionnaire to a significant group of the population, so the results are relevant, although there are some limitations.
Author comments: Thank you – appreciated
c) The references are numerous and current, including World Health Organization guides.
Author comments: Thank you – we wanted to provide an up-to-date discussion of key issues and challenges currently facing Iraq with respect to improving its antibiotic utilisation/ reducing AMR.

Reviewer 3 Report
Comments and Suggestions for Authors
This original article presents a cross-sectional study assessing knowledge, attitudes, and practices of Iraqi parents regarding antibiotic use in children. The topic is timely and addresses the global concern of antimicrobial resistance, particularly in a country where surveillance data is limited. The study is well-structured and provides useful insights for public health interventions and policy. However, a few methodological clarifications, terminology corrections, and language edits are necessary before it can be considered for publication.
- The study recruited participants from four major cities in Iraq. Please elaborate on the sampling strategy used (random, convenience, stratified) and how representativeness was ensured across different socioeconomic and educational strata. Clarification on how the sample size was increased (threefold) would also be beneficial.
- Although the questionnaire was developed based on previous literature and expert consultation, additional details regarding its reliability (Cronbach’s alpha) and validity (construct or content validity) are needed. Including the final version of the questionnaire as supplementary material would greatly enhance the manuscript's reproducibility.
- The study finds that university-educated and high-income parents are more likely to engage in self-medication with antibiotics. This is counterintuitive given that higher education is typically associated with better health literacy. Can the authors provide further discussion or hypotheses to explain this finding? For example, could overconfidence in self-diagnosis or easier access to antibiotics play a role?
- Did you perform any multivariable analyses (like logistic regression) to control for potential confounders? If not, please justify the choice of solely using Chi-square tests.
- Although limitations are briefly discussed, the manuscript would benefit from a more detailed section on potential biases, especially self-report bias, and how they might have influenced the results. Clarify any strategies implemented to mitigate these biases.
- Can you clarify if the data collection period (November 2023–September 2024) was completed as planned and if any deviations occurred?
Author Response
Quality of English Language
( ) The English could be improved to more clearly express the research.
(x) The English is fine and does not require any improvement.
Author comments: Thank you – appreciated
A) Comments and Suggestions for Authors
a) This original article presents a cross-sectional study assessing knowledge, attitudes, and practices of Iraqi parents regarding antibiotic use in children. The topic is timely and addresses the global concern of antimicrobial resistance, particularly in a country where surveillance data is limited. The study is well-structured and provides useful insights for public health interventions and policy. However, a few methodological clarifications, terminology corrections, and language edits are necessary before it can be considered for publication.
Author comments: Thank you for your kind comments and help. We hope we have adequately addressed the comments you made in the updated paper including additional references.
b) The study recruited participants from four major cities in Iraq. Please elaborate on the sampling strategy used (random, convenience, stratified) and how representativeness was ensured across different socioeconomic and educational strata. Clarification on how the sample size was increased (threefold) would also be beneficial.
Author comments: We appreciate the reviewer’s insightful comment and the opportunity to clarify our sampling strategy. We have now revised the Materials and Methods section (subsection 4.1) to elaborate on the sampling approach and justification for increasing the sample size. Specifically, we employed a stratified convenience sampling method to ensure broad representation across key demographic groups. The study was conducted in four major urban centres in Iraq (Kirkuk, Erbil, Tikrit, and Mosul), with equal representation (25%) from each city to ensure geographical balance. Within each city, participants were recruited from public locations such as shopping centres, parks, health facilities, and universities, targeting areas with high footfall to maximise socioeconomic diversity. This approach allowed for the inclusion of participants from a range of educational and income levels. To further ensure representativeness, we stratified recruitment efforts to capture variation across education (high school or lower vs. university graduate or higher) and income brackets (≤500,000 ID; 500,000–1,000,000 ID; >1,000,000 ID). The demographic characteristics of the sample reflected this diversity, as shown in Table 1.
Regarding the sample size: the minimum required sample was calculated using the Raosoft® sample size calculator, based on the combined population of the four cities (~4.37 million), a confidence level of 99%, and a 5% margin of error, yielding a required minimum of 664 participants. However, to enhance the statistical robustness of the analysis—particularly for subgroup comparisons based on education, income, and medical background—we increased the sample size by approximately threefold to 1,878 participants. This larger sample allowed for meaningful stratified analyses and subgroup comparisons.
We have now updated the text to reflect these additions, and trust this is now OK.
c) Although the questionnaire was developed based on previous literature and expert consultation, additional details regarding its reliability (Cronbach’s alpha) and validity (construct or content validity) are needed. Including the final version of the questionnaire as supplementary material would greatly enhance the manuscript's reproducibility.
Author comments: We appreciate the reviewer’s thoughtful and constructive comment. While we fully acknowledge the importance of assessing reliability and validity in questionnaire-based research, we respectfully note that the structure and purpose of our instrument did not necessitate the use of psychometric tests such as Cronbach’s alpha or construct validity assessment. Cronbach’s alpha is typically applied to assess the internal consistency of multiple items intended to measure a single latent construct (e.g., attitude or beliefs). Similarly, construct validity is commonly evaluated via factor analysis when a questionnaire includes scale-based items that are to be aggregated into composite scores. Our questionnaire, by contrast, was designed to elicit categorical, item-specific responses (e.g., Yes/No, Agree/Disagree) to independently assess parents’ knowledge, attitudes, and practices (KAP) regarding antibiotic use. Each item was analysed separately or in association with demographic variables, and not intended to form unidimensional scales or indices. As such, these psychometric techniques were not applicable to the analytical objectives of our study. That said, we did undertake measures to enhance the content and face validity of the tool. The questionnaire was developed following an extensive review of relevant literature and existing KAP instruments, and further refined through consultation with clinical pharmacy, epidemiology, and public health experts. It was also piloted with a group of parents to ensure clarity, relevance, and appropriateness of all items. We have now updated the paper to reflect this, and trust this is now acceptable.
In line with the reviewer’s helpful suggestion to enhance reproducibility and transparency, we have now also included the final version of the questionnaire as Supplementary File S1.
d) The study finds that university-educated and high-income parents are more likely to engage in self-medication with antibiotics. This is counterintuitive given that higher education is typically associated with better health literacy. Can the authors provide further discussion or hypotheses to explain this finding? For example, could overconfidence in self-diagnosis or easier access to antibiotics play a role?
Author comments: We thank the reviewer for this important and insightful observation. We agree that the finding appears counterintuitive and have now added further discussion in the revised manuscript to address this issue. While higher education is generally associated with improved health literacy, several studies among LMICs (now included) have shown that greater knowledge does not necessarily translate into appropriate behaviour, particularly in relation to antibiotic use. Behaviour is influenced by multiple factors beyond knowledge, including beliefs, social norms, convenience, access, perceived risk, and personal attitudes towards healthcare autonomy. One plausible explanation is that university-educated and higher-income individuals may demonstrate greater confidence in their ability to self-diagnose and manage illness, leading to increased self-medication. This phenomenon has been documented in other settings (now referenced). Another contributory factor may be easier access to antibiotics among higher-income families, either through purchasing power or social networks that enable bypassing formal healthcare channels. In some LMICs, better-off families may also have leftover antibiotics from previous prescriptions, facilitating their reuse. This paradox underscores the need for behavioural interventions that go beyond knowledge transfer, addressing underlying attitudes, risk perceptions, and self-efficacy. We have now included these hypotheses in the discussion to contextualise our findings, alongside incorporating additional references, and trust this is now acceptable.
e) Did you perform any multivariable analyses (like logistic regression) to control for potential confounders? If not, please justify the choice of solely using Chi-square tests.
Author comments: We thank the reviewer for this important question. In this study, we chose to employ Chi-square tests to examine associations between key demographic variables (e.g., education level, income, gender, medical background) and categorical outcomes relating to parents’ KAP toward antibiotic use.
While we recognise that multivariable analyses such as logistic regression can provide valuable insights into independent predictors by adjusting for potential confounding variables; however, we did not pursue these methods for several reasons:
1. Exploratory and Descriptive Aim: The primary aim of this study was to provide a descriptive and exploratory assessment of KAP patterns and to identify broad associations across strata. Our objective was not to develop predictive models but to generate baseline evidence for future hypothesis-driven research and pertinent targeted interventions to improve future knowledge and use of antibiotics. In such exploratory studies, bivariate analyses are frequently appropriate and informative (Sullivan & Artino, 2013).
2. Clarity and Transparency: We also wanted to ensure clarity and accessibility of the findings for a broad audience of public health professionals and policy-makers in Iraq, many of whom may be less familiar with multivariable statistical modelling. Presenting simple, stratified results by key variables we believe ensures transparent communication of results and supports stakeholder engagement (Reeves et al., 2008).
That said, we fully agree that multivariable modelling can be valuable in future studies, especially those seeking to assess causal inferences or develop targeted interventions. As such, we consider our findings as a foundation upon which we can build more complex analytical research and analysis. We have now updated our paper accordingly, including additional references, and trust this is now OK.
f) Although limitations are briefly discussed, the manuscript would benefit from a more detailed section on potential biases, especially self-report bias, and how they might have influenced the results. Clarify any strategies implemented to mitigate these biases.
Author comments: We thank the reviewer for highlighting this important aspect. We agree that a more detailed discussion of potential biases, particularly self-report bias, is warranted and we have now expanded the relevant section in the Discussion. As a cross-sectional survey rely on self-reported data, we accept that this study is susceptible to social desirability bias and recall bias, which may have led participants to under-report inappropriate behaviours (e.g., self-medication or early discontinuation of antibiotics) or over-report favourable behaviours. We attempted to minimise these risks through several measures. These included (i) ensuring anonymity and confidentiality of responses; (ii) using a structured questionnaire developed from previously validated tools; and (iii) conducting face-to-face interviews to clarify any ambiguities while avoiding influence on the content of responses. These steps have been shown in prior research to help reduce self-reporting bias in health surveys (Althubaiti, 2016). Nonetheless, we acknowledge that some residual bias may remain, and this is now more explicitly stated in the revised manuscript. We hope these changes, including additional references, are now acceptable.
g) Can you clarify if the data collection period (November 2023–September 2024) was completed as planned and if any deviations occurred?
Author comments: We thank the reviewer for this query. We confirm that data collection was completed as planned between November 2023 and September 2024, with no deviations from the proposed timeline. The process proceeded smoothly across the four targeted cities, with participant recruitment and data entry completed within the scheduled period. This consistency strengthens the internal validity of the study and ensures that the data reflect a coherent temporal snapshot of parental knowledge, attitudes, and practices during that period. We have added a statement in the discussion to reflect this, and trust this is now acceptable.
